# SPREAD SPURIOUS ATTRIBUTE: IMPROVING WORST-GROUP ACCURACY WITH SPURIOUS ATTRIBUTE ESTIMATION

**Junhyun Nam**[1], **Jaehyung Kim**[1], **Jaeho Lee**[2*], **Jinwoo Shin**[1]
[1]KAIST, [2]POSTECH
{junhyun.nam,jaehyungkim,jinwoos}@kaist.ac.kr
jaeho.lee@postech.ac.kr

## ABSTRACT

The paradigm of worst-group loss minimization has shown its promise in avoiding to learn spurious correlations, but requires costly additional supervision on spurious attributes. To resolve this, recent works focus on developing weaker forms of supervision—e.g., hyperparameters discovered with a small number of *group-labeled samples* with spurious attribute annotation—but none of the methods retain comparable performance to methods using full supervision on the spurious attribute. In this paper, instead of searching for weaker supervisions, we ask: Given access to a fixed number of group-labeled samples, what is the best achievable worst-group loss if we "fully exploit" them? To this end, we propose a pseudo-attribute-based algorithm, coined Spread Spurious Attribute (SSA), for improving the worst-group accuracy. In particular, we leverage samples both with and without spurious attribute annotations to train a model predicting the spurious attribute, then use the pseudo-attribute predicted by the trained model as a supervision on the spurious attribute to train a new robust model having minimal worst-group loss. Our experiments on various benchmark datasets show that our algorithm consistently outperforms the baseline methods using the same number of group-labeled samples. We also demonstrate that the proposed SSA can achieve comparable performances to methods using full (100%) spurious attribute supervision, by using a much smaller number of group-labeled samples—from 0.6% and up to 1.5%, depending on the dataset.

## 1 INTRODUCTION

Machine learning models trained on datasets containing spurious correlation (also known as "shortcuts") often end up learning such shortcuts instead of intended solutions (Geirhos et al., 2020). For example, consider an image classification dataset of 'cows' and 'camels,' in which most images of cows appear on grasslands and camels on deserts. When trained on such a dataset, models often learn to make predictions based on the landscape instead of the object (Beery et al., 2018). This phenomenon can lead to very low test accuracies on groups underrepresented in the training set.

Various approaches have been proposed to resolve this gap, and the idea of *minimizing the worst-group loss*—e.g., Sagawa et al. (2020)—has arisen as one of the most promising solutions. This approach forces high performances on both the majority group (e.g., cows standing on grasslands) and the minority group which contradicts the spurious correlation (e.g., cows standing on deserts). Despite its effectiveness, the worst-group loss minimization approach has a drawback: The learner requires supervision on which group each training sample belongs. For example, the learner needs to know that sample belongs to both the 'cow' and the 'desert' categories to utilize such group information to perform worst-group loss minimization. Even if we put aside the issue of identifying the spuriously correlated attributes ('desert' and 'grass') in the first place, one still needs to collect additional annotation on such spurious attributes. Acquiring such fine-grained annotation is presumably more expensive to collect, as the annotator needs a clear understanding of both the target attribute ('cow' and 'camel') and the spurious attribute ('desert' and 'grass').

---

*Work done at KAIST

Acknowledging this difficulty, recent works propose worst-group loss minimization algorithms that require a smaller number of group-labeled training samples (i.e., training samples with spurious attribute annotations) (Nam et al., 2020; Liu et al., 2021). At a high level, these works share a similar strategy (Fig. 1, Left): The methods first use a specialized mechanism to identify minority group samples among *group-unlabeled training samples,* and train a model in a way that puts more emphasis on identified-as-minority samples, e.g., by upweighting. A small set of group-labeled samples are used to tune hyperparameters of this procedure; as Liu et al. (2021) shows, the performance of trained models are very sensitive to these hyperparameters, indicating the high dependency of such algorithms on the availability of the group-labeled samples. Although these methods achieve higher worst-group accuracy than completely annotation-free approaches, e.g., Sohoni et al. (2020), they fail to perform comparably to the algorithms which use full annotations on spurious attributes, e.g., Sagawa et al. (2020). This performance gap gives rise to the following question: Can we closely achieve the full-annotation performance using a partially annotated dataset, if we use group-labeled samples more *actively* than hyperparameter-tuning?

**Contribution.** This paper proposes a worst-group loss minimization algorithm, coined Spread Spurious Attribute (SSA). At a high level, SSA consists of two phases—*pseudo-labeling* and *robust training*—that are designed to make a full use out of the group-labeled samples (Fig. 1, Right):

- *Pseudo-labeling:* SSA trains a spurious attribute predictor, using both group-labeled and group-unlabeled samples (i.e., samples lacking spurious attribute annotations). We find that group imbalances underlying the dataset can render pseudo-labels to be biased towards the majority group, which can be detrimental to the performance (as in Kim et al. (2020)), and propose using *group-wise adaptive thresholds* to mitigate this problem. The predictions are used as pseudo-labels (Lee, 2013) on the group-unlabeled samples.
- *Robust training:* Based on the pseudo-labeled training set, SSA trains a model which predicts the target attribute, using the worst-group loss minimization algorithms developed for fully supervised cases. We use Group DRO (Sagawa et al., 2020) as a default choice, and re-use group-labeled samples for the hyperparameter tuning.

Our experimental results suggest that SSA is a general yet effective framework for worst-group loss minimization. On various benchmark datasets, SSA consistently achieves performances comparable to fully supervised Group DRO and outperforms other baseline methods even when using only 5% of the group-labeled samples compared to the baselines. To be more specific, SSA requires less than 1.5% (as little as 0.6%) of group-labeled samples—993 out of 182637 on CelebA, and 4123 out of 288637 on MultiNLI—to achieve a worst-group accuracy similar to that of 100% usage. Moreover, we find that such benefits of SSA persist when combined with other robust training methods, such as Correct-N-Contrast (CNC[1]; Zhang et al. (2021)), the supervised contrastive learning with contrastive batch sampling using group information. Finally, we also empirically observe that the group-wise adaptive thresholding technique—which we proposed for better pseudo-labeling—enjoy a broader usage for addressing the general problem of semi-supervised learning under class imbalance; existing pseudo-labeling techniques (Kim et al., 2020; Wei et al., 2021) require certain assumptions in datasets to work well for the task, while ours does not.

## 2 RELATED WORKS

**Improving worst-group accuracy with group annotations.** It has been known in various literature that machine learning models often perform significantly worse on the samples from groups with a relatively small number of training samples than on samples from the majority group. The *class imbalance* problem (Japkowicz, 2000; Johnson & Khoshgoftaar, 2019) is one of the representative cases of this phenomenon. Here, class labels naturally define group identities and thus do not require any additional group annotations. In this context, popular strategies to improve worst-group performances are using *re-weighted loss* for each group (Huang et al., 2016; Khan et al., 2017) and *re-sampling* the given dataset to balance the group distribution of the training dataset (Chawla et al., 2002; He & Garcia, 2009). This paper, in contrast, considers a setup where the group is defined as a combination of the label (target attribute) and the spuriously correlated attribute; unlike the class imbalance literature, the availability of the full group identity information is not guaranteed,

---

[1]We use the term CNC to refer to its supervised contrastive learning procedure with contrastive batch sampling only, rather than the entire process which includes spurious attribute annotation estimation via ERM.

**Objective**: classify ◯ vs. □
**Obstacle**: color and shape are highly correlated (most of ◯ are 🟡, □ are 🔵)
no color annotation for the training set

Figure 1: Comparison of prior approaches (left) and the proposed SSA (right). **Left:** Prior works use group-labeled samples only for hyperparameter tuning. **Right:** The proposed SSA uses group-labeled samples for both training and validation of the model.

as spurious attributes may not have proper annotations. Under the setting where spurious attribute annotations are available on all samples, Sagawa et al. (2020) gives an online optimization algorithm that shows promising results on minimizing the worst-group loss.

**Improving worst-group accuracy without group annotations.** To reduce additional annotation costs, recent works aim to train a robust model without requiring group annotations for all training samples. These works utilize models trained with standard procedure to identify samples that disagree with spurious correlations. Nam et al. (2020) train a model to be intentionally biased using the generalized cross entropy loss, and use it to identify-and-upweight high-loss samples for training another model. Liu et al. (2021) train a standard ERM model for a few epochs and upweight samples misclassified by this model to train the second model. Although these approaches do not use any group information for training, they still use a small number of group-labeled samples for hyperparameter tuning, which is critical to their worst-group performance. In this paper, we design a method to utilize group-labeled samples more efficiently. We note that, in the class imbalance literature, another line of works proposes to use group-labeled samples more actively; they use labeled samples for determining sample weights for the training set through meta-learning (Ren et al., 2018) or training a model to predict sample weights (Shu et al., 2019).

## 3 PROBLEM SETUP

Consider the learning a classifier in the presence of spurious correlations in the training set. Following the prior work of Sagawa et al. (2020), we cast this problem as minimizing the worst-group loss, where the group identity is determined by the target attribute (that we want to predict) and the spurious attribute (that we want to ignore). We assume that we do not have spurious attribute annotations of the samples in the training set (*group-unlabeled set*), but have an access to additional set of samples with both spurioust attribute and target attribute annotations (*group-labeled set*).

More formally, we let each sample be a triplet consisting of an *input* $x \in \mathcal{X}$, a *target attribute* $y \in \mathcal{Y}$, and a *spurious attribute* $a \in \mathcal{A}$. Our goal is to train a parameterized model $f_\theta : \mathcal{X} \to \mathcal{Y}$ that minimizes the worst-group expected loss on test samples; for the purpose of avoiding learning spurious correlations, we define the *group* as an attribute pair $g := (y, a) \in \mathcal{Y} \times \mathcal{A} =: \mathcal{G}$. In other words, we aim to minimize

$$R(\theta) = \max_{g \in \mathcal{G}} \mathbb{E}_{(x,y,a) \sim P_g}[\ell(f_\theta(x), y)], \tag{1}$$

for some loss function $\ell : \mathcal{Y} \times \mathcal{Y} \to \mathbb{R}$, where $P_g$ denotes the group-conditioned data-generating distribution. Note that, as we focus on the group defined by the attribute pairs, $P_g$ is deterministic on $y$ and $a$, and stochastic only on $x$.

We assume that the learner has access to two types of samples for training: The group-unlabeled set consists of $n$ samples without spurious attribute annotations, i.e., $\mathcal{D}_U := \{(x_1, y_1), \ldots, (x_n, y_n)\}$, and the group-labeled set consists of $m$ samples with spurious attribute annotations, i.e., $\mathcal{D}_L := \{(\tilde{x}_1, \tilde{y}_1, \tilde{a}_1), \ldots, (\tilde{x}_m, \tilde{y}_m, \tilde{a}_m)\}$. We assume that the group-labeled samples are much more costly to collect, and thus we have $n \gg m$.

## 4 SPREAD SPURIOUS ATTRIBUTE

We now describe the algorithm we propose, Spread Spurious Attribute (SSA). At a high level, SSA consists of two phases: *pseudo-labeling* and *robust training*. In the pseudo-labeling phase, SSA trains a model to predict the spurious attribute. In particular, we use *both* group-labeled and group-unlabeled samples to generate pseudo-labels on the group-unlabeled training samples (Section 4.1), with an adaptive thresholding technique for balancing the number of samples in each group (Section 4.2). Then, in the robust training phase, SSA uses generated pseudo-labels to train a model that predicts the target attribute with a small worst-group loss (Section 4.3). We use Group DRO (Sagawa et al., 2020) as our default robust training method, but SSA also performs well when combined with other existing algorithms that use full spurious label supervisions (see Section 5.3).

### 4.1 PSEUDO-LABELING USING GROUP-LABELED AND GROUP-UNLABELED SAMPLES

Using both group-labeled set $\mathcal{D}_L$ and group-unlabeled set $\mathcal{D}_U$, we train a model to predict spurious attributes on the group-unlabeled samples. The predictions will be used to generate artificial spurious attribute labels on the group-unlabeled training samples, called pseudo-labels (Lee, 2013). We emphasize that we also use $\mathcal{D}_U$ for *training* the model, instead of training solely based on $\mathcal{D}_L$. In fact, as we shall see in Section 5.2, the additional use of training set brings a significant performance boost. More specifically, we first partition both the group-labeled and group-unlabeled set into two:

$$\mathcal{D}_L = \mathcal{D}_L^\circ \cup \mathcal{D}_L^\bullet, \quad \mathcal{D}_U = \mathcal{D}_U^\circ \cup \mathcal{D}_U^\bullet. \tag{2}$$

We use $\mathcal{D}_L^\circ, \mathcal{D}_U^\circ$ to *train* the spurious attribute predictor that make prediction on $\mathcal{D}_U^\bullet$, and validate the model with $\mathcal{D}_L^\bullet$.[2] To train this predictor, we use a loss function consisting of two terms: the supervised loss for the samples in $\mathcal{D}_L^\circ$, and the unsupervised loss for samples in $\mathcal{D}_U^\circ$. The supervised loss is simply the standard cross entropy loss. For the unsupervised loss, we use the cross entropy loss between the prediction and the *pseudo-labels*, i.e., labels generated by taking the $\arg\max$ of predictions. Following prior works (e.g., Sohn et al. (2020)), we apply the loss only if the confidence of the prediction exceeds some threshold $\tau \geq 0$. More formally, let us denote the class probability estimate of the predictor on the attribute $a$ given the input $x$ as $\hat{p}(a|x)$, and the pseudo-label from this prediction as $\hat{a}(x) = \arg\max_{a \in \mathcal{A}} \hat{p}(a|x)$. Then, the supervised and unsupervised losses are

$$\ell_{\mathrm{sup}}(x, y, a) = \mathrm{CE}(\hat{p}(\cdot|x), a), \quad \ell_{\mathrm{unsup}}(x, y) = \mathbf{1}\left\{ \max_{a \in \mathcal{A}} \hat{p}(a|x) \geq \tau \right\} \cdot \mathrm{CE}(\hat{p}(\cdot|x), \hat{a}(x)), \tag{3}$$

where $\mathrm{CE}(\hat{p}(\cdot|x), a)$ denotes the cross-entropy loss between the prediction $\hat{p}(\cdot|x)$ and label $a$. The total loss is then given as the sum of the supervised and unsupervised loss

$$\mathcal{L} = \mathbb{E}_{\mathcal{D}_L^\circ}[\ell_{\mathrm{sup}}(x, y, a)] + \mathbb{E}_{\mathcal{D}_U^\circ}[\ell_{\mathrm{unsup}}(x, y)] \tag{4}$$

As we will describe in Section 4.2, we additionally use group-wise adaptive thresholds for pseudo-labeling (i.e., set different $\tau$ for each group). The thresholds are determined in a way that balances the *pseudo-group* (i.e., the pair $\hat{g} = (y, \hat{a}(x))$) population of the samples with confidence exceeding the group-wise threshold. This strategy helps the model to avoid making a pseudo-label prediction biased toward the majority group. Also, we note that we make predictions only on samples in $\mathcal{D}_U^\bullet$ and not on samples in $\mathcal{D}_U^\circ$; empirically, we observe that this "splitting" of group-unlabeled samples is beneficial for performance comparing to using the whole $\mathcal{D}_U$ for training (at the cost of running pseudo-labeling multiple times). For more discussions and ablation studies, see Appendix A.2.

### 4.2 BALANCING GENERATED PSEUDO-LABELS VIA ADAPTIVE THRESHOLDS

As we train the spurious attribute predictor with the loss (Eq. (4)), the prediction confidence of the model on each sample gradually increases. Ideally, we want the number of highly-confident samples (i.e., with confidence greater than $\tau$) to increase uniformly over all pseudo-groups so that each pseudo-group contributes evenly to weight updates. However, if the underlying group population is severely imbalanced—as is common with spurious attributes—samples from the majority group often attains high prediction confidence significantly faster than samples from minority groups, by receiving more frequent gradient updates. This training imbalance leads to a severer imbalance in the pseudo-labels, resulting in detrimental effects on the downstream training.

---

[2]We write $\circ$ to denote that samples are "visible" during the training phase, and $\bullet$ to denote that they are not.

To mitigate this majority group bias, we set different thresholds for each pseudo-group, so that the same number of samples from each group is used for training with Eq. (4). To do this, we first set a fixed threshold $\tau_{g_{\min}}$ for the *group with the smallest population* in the training split of the group-labeled set, i.e., the group defined as

$$g_{\min} = \underset{g \in \mathcal{Y} \times \mathcal{A}}{\arg \min} |\mathcal{D}_L^\circ(g)|, \qquad \text{where } \mathcal{D}_L^\circ(g) := \left\{ (x, y, a) \in \mathcal{D}_L^\circ \mid g = (y, a) \right\}. \tag{5}$$

Next, we count the number of samples in $\mathcal{D}_U^\circ$ which (a) belongs to $g_{\min}$ after pseudo-labeling, and (b) the prediction confidence exceeds $\tau_{g_{\min}}$. Then, we set thresholds for other groups to have same number of samples from each group. Concretely, let $\mathcal{D}_U^\circ(g, \tau_g)$ be the set of samples in the training split of the group-unlabeled set with pseudo-group $g$ and confidence greater than equal to $\tau_g$, i.e.,

$$\mathcal{D}_U^\circ(g, \tau_g) := \left\{ (x, y) \in \mathcal{D}_U^\circ \mid g = (y, \hat{a}(x)), \quad \max_{a \in \mathcal{A}} \hat{p}(a|x) \geq \tau_g \right\}. \tag{6}$$

Then, for each group $g \neq g_{\min}$, we set $\tau_g$ to be the smallest real number such that

$$|\mathcal{D}_L^\circ(g)| + |\mathcal{D}_U^\circ(g, \tau_g)| \leq |\mathcal{D}_L^\circ(g_{\min})| + |\mathcal{D}_U^\circ(g_{\min}, \tau_{g_{\min}})|. \tag{7}$$

With this group-wise adaptive threshold, we use the unsupervised loss revised as

$$\ell_{\text{unsup}}(x, y) = \mathbf{1} \left\{ \max_{a \in \mathcal{A}} \hat{p}(a|x) \geq \tau_{(y, \hat{a}(x))} \right\} \cdot \text{CE} \left( \hat{p}(\cdot|x), \hat{a}(x) \right). \tag{8}$$

We show the effectiveness of this group-wise adaptive threshold in Section 5.2, and further applicability to class-imbalance problem in Section 5.4.

### 4.3 WORST-GROUP LOSS MINIMIZATION WITH ESTIMATED PSEUDO-GROUPS

After training the model using the revised loss (Section 4.2), we generate final pseudo-labels for all samples in the training set. In other words, we generate

$$\widetilde{\mathcal{D}}_U = \left\{ (x, y, \hat{a}(x)) \mid (x, y) \in \mathcal{D}_U \right\}. \tag{9}$$

We put pseudo-labels on all samples without applying any threshold, which allows us to utilize the whole training set. In the robust training phase, we use the pseudo-labeled dataset $\widetilde{\mathcal{D}}_U$ and the group-labeled dataset $\mathcal{D}_L$ to perform worst-group loss minimization. We use Group DRO (Sagawa et al., 2020) as our default algorithm, using $\widetilde{\mathcal{D}}_U$ for training and $\mathcal{D}_L$ for validation. We note that SSA can also use other robust training subroutines instead of group DRO; in Section 5.3, we show that our framework performs well when combined with CNC (Zhang et al., 2021).

## 5 EXPERIMENTS

Here, we briefly describe the experiment setup that will be used throughout all experiments in this section, except for Section 5.4 where we consider a slightly different scenario.

**Datasets.** We evaluate SSA on two image classification datasets (Waterbirds, CelebA) and two natural language processing datasets (MultiNLI, CivilComments-WILDS) containing spurious correlations. For all datasets, we use the *validation* split of the dataset as the group-labeled set. Below, we briefly describe each dataset and the corresponding spurious correlations.

- *Waterbirds* (Sagawa et al., 2020): Waterbirds is an artificial dataset generated by combining bird photographs in the Caltech-UCSD Birds dataset (Wah et al., 2011) with landscapes from Places (Zhou et al., 2017). The goal is to classify the target attributes $\mathcal{Y} = \{\text{waterbird, landbird}\}$ given the spurious correlations with the background landscape $\mathcal{A} = \{\text{water background, land background}\}$.
- *CelebA* (Liu et al., 2015): CelebA dataset consists of the face pictures of celebrities, with various annotations on facial/demographic features. We use the hair color as the target attribute $\mathcal{Y} = \{\text{blond, non-blond}\}$, given the spurious correlations with the gender $\mathcal{A} = \{\text{male, female}\}$.
- *MultiNLI* (Williams et al., 2018): MultiNLI is a multi-genre natural language corpus where each data instance consists of two sentences and a label indicating whether the second sentence is entailed by, contradicts, or neutral to the first. We use this label as the target attribute (i.e., $\mathcal{Y} = \{\text{entailed, neutral, contradictory}\}$), and use the existence of the negating words as the spurious attribute (i.e., $\mathcal{A} = \{\text{negation, no negation}\}$).

Table 1: Average and worst-group test accuracies evaluated on image classification datasets (Waterbirds, CelebA). For more rigorous comparison, we run SSA and Group DRO on 3 random seeds and report the average and the standard deviation. Results of ERM, CVaR DRO, LfF and JTT are from Liu et al. (2021). Results of EIIL on Waterbirds are from Creager et al. (2021). Best performances (among methods using only validation set labels) are marked in bold.

| Method | Amount of group label used | Waterbirds | | CelebA | |
|---|---|---|---|---|---|
| | | Avg. | Worst-group | Avg. | Worst-group |
| ERM | val. set | 97.3 | 72.6 | 95.6 | 47.2 |
| CVaR DRO (Levy et al., 2020) | val. set | 96.0 | 75.9 | 82.5 | 64.4 |
| LfF (Nam et al., 2020) | val. set | 91.2 | 78.0 | 85.1 | 77.2 |
| EIIL (Creager et al., 2021) | val. set | 96.9 | 78.7 | 91.9 | 83.3 |
| JTT (Liu et al., 2021) | val. set | 93.3 | 86.7 | 88.0 | 81.1 |
| SSA (Ours) | val. set | $92.2_{\pm 0.87}$ | $\mathbf{89.0}_{\pm 0.55}$ | $92.8_{\pm 0.11}$ | $\mathbf{89.8}_{\pm 1.28}$ |
| | 5% of val. set | $92.6_{\pm 0.15}$ | $87.1_{\pm 0.70}$ | $92.8_{\pm 0.34}$ | $86.7_{\pm 1.11}$ |
| Group DRO (Sagawa et al., 2020) | train. & val. set | $91.8_{\pm 0.48}$ | $89.2_{\pm 0.18}$ | $93.1_{\pm 0.21}$ | $88.5_{\pm 1.16}$ |

Table 2: Average and worst-group test accuracies evaluated on natural language datasets (MultiNLI, CivilComments-WILDS). For more rigorous comparison, we run SSA and Group DRO on 3 random seeds and report the average and the standard deviation. Results of ERM, CVaR DRO, LfF and JTT are from Liu et al. (2021). Results of EIIL on CivilComments are from Creager et al. (2021). Best performances (among methods using only validation set labels) are marked in bold.

| Method | Amount of group label used | MultiNLI | | CivilComments-WILDS | |
|---|---|---|---|---|---|
| | | Avg. | Worst-group | Avg. | Worst-group |
| ERM | val. set | 82.4 | 67.9 | 92.6 | 57.4 |
| CVaR DRO (Levy et al., 2020) | val. set | 82.0 | 68.0 | 92.5 | 60.5 |
| LfF (Nam et al., 2020) | val. set | 80.8 | 70.2 | 92.5 | 58.8 |
| EIIL (Creager et al., 2021) | val. set | 79.4 | 70.9 | 90.5 | 67.0 |
| JTT (Liu et al., 2021) | val. set | 78.6 | 72.6 | 91.1 | 69.3 |
| SSA (Ours) | val. set | $79.9_{\pm 0.87}$ | $\mathbf{76.6}_{\pm 0.66}$ | $88.2_{\pm 1.95}$ | $\mathbf{69.9}_{\pm 2.02}$ |
| | 5% of val. set | $80.4_{\pm 0.62}$ | $76.5_{\pm 1.89}$ | $89.1_{\pm 1.09}$ | $69.5_{\pm 1.15}$ |
| Group DRO (Sagawa et al., 2020) | train. & val. set | $81.4_{\pm 1.40}$ | $76.6_{\pm 0.41}$ | $87.7_{\pm 1.35}$ | $69.1_{\pm 1.53}$ |

- *CivilComments-WILDS* (Borkan et al., 2019; Koh et al., 2021): CivilComments-WILDS consists of comments generated by online users, each of which are labeled with the toxicity indicator $\mathcal{Y} = \{\text{toxic}, \text{non-toxic}\}$. We use demographic attributes of the mentioned identity $\mathcal{A} = \{\text{male}, \text{female}, \text{White}, \text{Black}, \text{LGBTQ}, \text{Muslim}, \text{Christian}, \text{other religion}\}$ as a spurious attribute for evaluation purpose. We note that a comment can contain multiple such identities, so that groups defined by $\mathcal{Y} \times \mathcal{A}$ can be overlapped. Therefore, we use $\mathcal{A}' = \{\text{any identity}, \text{no identity}\}$ as a spurious attribute for training, following Liu et al. (2021).

**Models.** For the all experiments on image classification datasets, we use ResNet-50 (He et al., 2016) starting from ImageNet-pretrained weights. For experiments on language datasets, we use pretrained BERT (Devlin et al., 2019). We use the same architecture for predicting the spurious attribute (in the pseudo-labeling phase) and the target attribute (in the robust training phase).

## 5.1 MAIN RESULTS

We compare the average and worst-case performance of the proposed SSA against the standard empirical risk minimization (ERM) and recent methods that tackles spurious correlation without group annotation for the training, including CVaR DRO (Levy et al., 2020), LfF (Nam et al., 2020), EIIL (Creager et al., 2021), JTT (Liu et al., 2021), and Group DRO (Sagawa et al., 2020) requiring group annotation to minimize the worst-group loss.

In Table 1, 2, we report average accuracies and the worst-group accuracies on all datasets we consider. Our method consistently outperforms all the other approaches that use spurious attribute annotated dataset for validation while using the same amount of spurious attribute annotation. Notably, our method shows comparable performance to Group DRO which uses full amount of spurious attribute annotation for the training set, even outperforms on CelebA dataset. We also run our algo-

Table 3: Worst-group accuracy on Waterbirds and CelebA with varying group-labeled set size. Results of JTT are from Liu et al. (2021).

| Method | Waterbirds | | | | CelebA | | | |
|---|---|---|---|---|---|---|---|---|
| | 100% | 20% | 10% | 5% | 100% | 20% | 10% | 5% |
| JTT (Liu et al., 2021) | 86.7 | 84.0 | 86.9 | 76.0 | 81.1 | 81.1 | 81.1 | 82.2 |
| SSA (Ours) | 89.0 | 88.9 | 88.9 | 87.1 | 89.8 | 88.9 | 90.0 | 86.7 |

Table 4: Group-wise accuracy (recall) of spurious attribute predictor trained in the pseudo-labeling phase, on the CelebA dataset. Worst-group accuracies are marked in bold.

| Method | Amount of group label used | # (Blond, Male) in $\mathcal{D}_L$ | Non-blond | | Blond | |
|---|---|---|---|---|---|---|
| | | | Female | Male | Female | Male |
| Vanilla | | | 90.2 | 90.3 | 97.0 | **70.4** |
| Pseudo-labeling | 10% of val. set | 18 | 94.2 | 95.5 | 98.8 | **76.4** |
| + Group-wise threshold | | | 89.5 | 93.9 | 95.1 | **83.6** |
| Vanilla | | | 88.0 | 89.7 | 96.3 | **68.7** |
| Pseudo-labeling | 5% of val. set | 8 | 92.2 | 94.2 | 98.4 | **67.8** |
| + Group-wise threshold | | | 83.3 | 90.6 | 92.7 | **75.7** |

rithm using only 5% of the default validation set to show efficiency of our algorithm to improve the worst-group accuracy. We further provide analysis on varying group-labeled set size below.

**Effect of group-labeled set size.** Although we focus on improving the worst-group performance with given amount of spurious annotation, reducing the amount of supervision is an important topic to discuss especially with high annotation cost. In the main results in Table 1, 2, we use the default validation sets provided by each dataset as $\mathcal{D}_L$. To further test whether SSA can achieve high worst-group accuracy with reduced amount of supervision, we run our algorithm with small fraction of the default validation sets as group-labeled sets. Following Liu et al. (2021), we run our method using 100%, 20%, 10%, 5% of the default validation set. In Table 3, we report the worst-group accuracy of JTT and our algorithm on Waterbirds and CelebA, with various group-labeled set size. Surprisingly, we find that our method maintains high worst-group accuracy even with the 10% of the original validation set. Most notably, the number of attribute annotated samples used for training spurious attribute predictor is 58 in total, 6 for the worst-group in Waterbirds when we only use 10% of the default validation set.

## 5.2 DETAILED ANALYSIS ON THE PSEUDO-LABELING PHASE OF SSA

We now take a closer look at the pseudo-labeling phase of the proposed SSA. In particular, we focus on validating the effectiveness of the group-wise adaptive threshold we introduced in Section 4.2. The purpose of the adaptive threshold was to prevent the pseudo-labels from being biased towards the majority group during the pseudo-labeling phase; the prediction confidence of the majority group samples may exceed the threshold faster than samples of minority group, increasing the contribution of majority-group samples on the loss even further. In the first set of experiments (Table 4), we perform ablation studies on the pseudo-labeling and the adaptive threshold to see their effects on the spurious attribute prediction performance of the SSA. In the second set of experiments (Table 5), we validate if the pseudo-labels are biased towards the majority group and check that our adaptive threshold strategy successfully addresses the phenomenon.

**Accuracy of the spurious attribute predictor.** In Table 4, we report the group-wise spurious attribute prediction accuracies of SSA on CelebA dataset (using 5% or 10% of the validation split as the group-labeled set) in pseudo-labeling phase for following ablations: (1) *Vanilla*: Does not use any pseudo-labels and train the model using only the validation set samples. (2) *Pseudo-labeling*: Uses pseudo-labeling with a fixed threshold, and (3) *+Group-wise threshold*: Identical to SSA, using group-wise adaptive thresholds for pseudo-labeling. We find that using the group-wise threshold increases the spurious attribute prediction accuracy for the worst-group—$(y, a) = (\text{Blond}, \text{Male})$, providing ~7% boost in both cases. Interestingly, we observe that pseudo-labeling with fixed thresholds can even slightly degrade the performance, when the validation set is too small (5%).

Table 5: Samples statistics during the training of spurious attribute predictor on CelebA with 5% of the default validation set.

| Description | Non-blond | | Blond | | Fraction of |
| --- | --- | --- | --- | --- | --- |
| | Female | Male | Female | Male | (Blond, Male) |
| Number of group-labeled samples ($|\mathcal{D}_L^\circ|$) | 213 | 206 | 71 | 4 | 0.81% |
| Number of group-unlabeled samples ($|\mathcal{D}_U^\circ|$) | 47750 | 44617 | 15240 | 906 | 0.83% |
| Pseudo-labeling | | | | | |
| Number of samples exceeding fixed threshold | 45897 | 43201 | 15344 | 712 | 0.68% |
| Number of samples used for training | 45897 | 43201 | 15344 | 712 | 0.68% |
| + Group-wise threshold | | | | | |
| Number of samples exceeding fixed threshold | 32926 | 34622 | 12733 | 712 | 0.88% |
| Number of samples used for training | 501 | 528 | 681 | 716 | 29.5% |

Table 6: Average and the worst-group accuracy of the proposed SSA algorithm, when combined with two different choices of supervised worst-case loss minimization algorithms. The numbers in brackets are the gap closed by SSA between ERM and supervised robust training methods.

| Robust training method | Group label | Waterbirds | | CelebA | |
| --- | --- | --- | --- | --- | --- |
| | | Avg. | Worst-group | Avg. | Worst-group |
| ERM | | 97.3 | 72.6 | 95.6 | 47.2 |
| Group DRO (Sagawa et al., 2020) | Ground truth | $91.8_{\pm0.48}$ | $89.2_{\pm0.18}$ | $93.1_{\pm0.21}$ | $88.5_{\pm1.16}$ |
| | SSA (Ours) | $92.2_{\pm0.87}$ | $89.0_{\pm0.55}$ | $92.8_{\pm0.11}$ | $89.8_{\pm1.28}$ |
| Gap closed by SSA | | | (98.8%) | | (100%) |
| CNC (Zhang et al., 2021) | Ground truth | $92.5_{\pm0.49}$ | $90.0_{\pm0.56}$ | $92.5_{\pm0.62}$ | $87.4_{\pm0.85}$ |
| | SSA (Ours) | $92.6_{\pm0.40}$ | $89.2_{\pm0.24}$ | $91.7_{\pm1.32}$ | $87.6_{\pm0.32}$ |
| Gap closed by SSA | | | (95.4%) | | (100%) |

**Population comparison.** In Table 5, we compare the populations of pseudo-labeled samples that contribute to the training, for pseudo-labeling using a fixed threshold ('Pseudo-labeling') and the adaptive threshold ('+Group-wise threshold'); for each method, we trained the spurious attribute predictor until the worst-group spurious attribute prediction accuracy reaches the highest point. We used CelebA dataset with only 5% of the validation split. From the experimental results, we find that naïve pseudo-labeling with a fixed threshold indeed leads the pseudo-labels to be biased towards the majority group using only 0.68% of the minority group (male blond) for training while the true fraction of the group is over 0.8%. On the other hand, using the adaptive threshold successfully addresses this problem, lifting the fraction of blond male samples to 0.88%, which is close to the population level. More impressively, we observe that pseudo-labeling phase adaptive threshold uses relatively uniform number of samples from each group to training the spurious attribute predictor.

## 5.3 SSA COMBINED WITH SUPERVISED CONTRASTIVE LEARNING

We now examine whether the proposed SSA still remains to be beneficial when combined with other robust training procedures. As an example, we choose a recent robust training procedure (Zhang et al., 2021) proposed as an alternative to Group DRO. More specifically, CNC (Zhang et al., 2021) considers a following procedure base on the supervised contrastive loss (Khosla et al., 2020): As in JTT (Liu et al., 2021), we first train a standard ERM model. Then, we use the contrastive loss to maximize the representational similarity between samples have the same target label but different ERM prediction, while minimizing the representational similarity of samples with different target attribute but same ERM prediction. This procedure can be smoothly combined the SSA framework, by replacing the ERM predictions with the pseudo-labels generated in the first phase of SSA.

In Table 6, we compare the performance of SSA using Group DRO or CNC with the performance of the robust training methods using the full spurious attribute annotations on the training set. Both models learned with SSA achieved comparable performance to the fully supervised counterparts. This result suggests that the benefit of SSA may not be constrained on a specific robust training method, and may be combined with more general classes of robust training algorithms. Also, we find that none of the robust training method consistently outperform the other; CNC with group label slightly outperforms Group DRO on Waterbirds, and Group DRO does better on CelebA.

Table 7: Comparison of classification performance (bACC/GM) on CIFAR-10 under 4 different sets of $(m_{\text{maj}}, n_{\text{maj}}, \gamma_{\text{lab}})$, where $m_{\text{maj}}$ denotes number of labeled samples in to the largest class, $n_{\text{maj}}$ denotes the number of unlabeled samples in the largest class, and $\gamma_{\text{lab}}$ denotes the largest-to-smallest class ratio in the labeled set (we let $\gamma_{\text{unlab}} = 1$). The best results are indicated in bold.

| Algorithm | $(m_{\text{maj}}, n_{\text{maj}}, \gamma_{\text{lab}})$ | | | |
| | $(500, 4500, 100)$ | $(500, 4500, 50)$ | $(100, 4900, 50)$ | $(100, 4900, 20)$ |
|---|---|---|---|---|
| Vanilla | $42.1_{\pm 0.66}$ / $25.5_{\pm 1.16}$ | $50.3_{\pm 0.14}$ / $41.6_{\pm 0.40}$ | $28.0_{\pm 0.97}$ / $13.0_{\pm 1.90}$ | $34.4_{\pm 1.78}$ / $26.4_{\pm 2.86}$ |
| FixMatch (Sohn et al., 2020) | $65.6_{\pm 0.35}$ / $26.8_{\pm 0.82}$ | $68.3_{\pm 0.15}$ / $30.4_{\pm 0.28}$ | $56.4_{\pm 0.23}$ / $14.9_{\pm 1.21}$ | $69.1_{\pm 1.96}$ / $42.1_{\pm 2.37}$ |
| DARP (Kim et al., 2020) | $75.6_{\pm 0.26}$ / $73.1_{\pm 0.19}$ | $78.2_{\pm 0.14}$ / $76.5_{\pm 0.26}$ | $72.3_{\pm 0.28}$ / $67.8_{\pm 0.45}$ | $77.5_{\pm 0.47}$ / $75.2_{\pm 0.71}$ |
| CReST (Wei et al., 2021) | $77.4_{\pm 0.36}$ / $61.6_{\pm 2.11}$ | $79.7_{\pm 1.00}$ / $77.2_{\pm 1.01}$ | $67.4_{\pm 0.70}$ / $34.7_{\pm 3.23}$ | $69.8_{\pm 0.35}$ / $53.6_{\pm 0.42}$ |
| Adaptive thresholds (Ours) | $\mathbf{86.1}_{\pm 0.44}$ / $\mathbf{85.7}_{\pm 0.47}$ | $\mathbf{87.4}_{\pm 0.03}$ / $\mathbf{87.1}_{\pm 0.02}$ | $\mathbf{83.8}_{\pm 0.19}$ / $\mathbf{82.7}_{\pm 0.43}$ | $\mathbf{86.5}_{\pm 0.16}$ / $\mathbf{86.1}_{\pm 0.16}$ |
| Oracle | $93.6_{\pm 0.18}$ / $93.5_{\pm 0.18}$ | $93.6_{\pm 0.18}$ / $93.5_{\pm 0.18}$ | $93.6_{\pm 0.15}$ / $93.6_{\pm 0.15}$ | $93.9_{\pm 0.02}$ / $93.8_{\pm 0.03}$ |

## 5.4 APPLICATION TO GENERAL SEMI-SUPERVISED LEARNING UNDER CLASS IMBALANCE

In Section 4.2, we proposed to use adaptive thresholding to mitigate confirmation bias when pseudo-labeling group-imbalanced datasets. Interestingly, it turns out that the benefit of this idea also extends (without any modification) to a more general scenario of semi-supervised learning (SSL) on datasets with class imbalances. In fact, two settings are quite similar, except that SSA aims to put pseudo-labels on spurious attributes while SSL methods estimate target attributes. To demonstrate this point, we evaluate adaptive thresholds under the SSL setup, and compare it with the performances of baseline pseudo-labeling-based SSL algorithms. FixMatch (Sohn et al., 2020) is a recent SSL method proposed without considerations on the class imbalance issue. DARP (Kim et al., 2020) and CReST (Wei et al., 2021) build on FixMatch to handle class imbalance, but are primarily designed under the assumptions that there exists sufficiently many labeled data at hand (to estimate the class imbalance ratio), and that class distributions of labeled and unlabeled datasets are identical, respectively.[3] In contrast, our method of adaptive thresholds does not rely on such assumptions to solve the same task.

We consider a slightly more challenging experimental setup than in Kim et al. (2020): We construct an artificial labeled dataset from the CIFAR-10 (Krizhevsky et al., 2009) by controlling the number of samples in the majority class $m_{\text{maj}}$ (i.e., the largest class) and the ratio between the largest and smallest class sizes $\gamma_{\text{lab}} \geq 1$.[4] In a similar manner, we construct an unlabeled dataset using some parameters $n_{\text{maj}}$ and $\gamma_{\text{unlab}}$. We select these parameters so that the number of labeled samples is small, and the imbalance ratios of labeled and unlabeled sets have bigger discrepancies (We provide further details in Appendix A.6). In Table 7, we observe that empirical gains from both DARP and CReST are limited comparing to supervised learning with full ground-truth labels (oracle), due to the violation of their inherent assumptions in the experimental setups of our choice, i.e., limited labeled data and different class distribution between labeled and unlabeled datasets. However, our method successfully reduces such gap by effectively constructing the pseudo-labels for minority classes. Overall, these results implies that the proposed method has a potential to provide a more robust semi-supervised learning solution (despite its simplicity) in more realistic scenarios, which we think is an interesting direction to explore further in the future.

## 6 CONCLUSION

In this work, we present Spread Spurious Attribute (SSA), an algorithm for improving worst-group accuracy in the presence of spurious correlation. SSA framework uses a small amount of spurious attribute annotated samples to estimate group identities of the training set samples. With the generated attribute annotated training set, we successfully train a robust model using existing worst-case loss minimization algorithms. SSA is highly effective given a limited amount of the spurious attribute annotated samples, but still does not completely remove the need for supervision on spurious attributes which is an important future direction.

---

[3] In fact, in the previous spurious attribute setups, adopting DARP/CReST methods did not provide much gain over naïve pseudo-labeling; we suspect the reason to be the violation of these assumptions.

[4] Larger $\gamma_{\text{lab}}$ thus indicates a more severe imbalance.

ETHICS STATEMENT

This paper addresses the problem of mitigating the harmful effects of spurious correlations in the dataset, which is deeply intertwined with the topics of machine bias and fairness. By targeting demographic groups (e.g., gender, race) carefully, our method has a potential to be used for preventing the machine to make predictions on the basis of biases on such demographic identities. One possible pitfall, however, is that this group-based evaluation can provide a false sense of morality; as making a fair decision in terms of group cannot be equated with the individual fairness (even putting aside the issues of *gerrymandering*), our performance reports based on the worst-group performance should not be naïvely taken as a *measure of justice.* About the responsible research practice: We have used benchmark datasets, and thus believe that our research practice has not posed any foreseeable hazard in this regard.

REPRODUCIBILITY STATEMENT

We provide descriptions of the experimental setup and implementation details in Appendices A.2, A.3 and A.5. Also, we provide our source code as a part of the open-to-public supplementary materials.

ACKNOWLEDGMENTS

This work was supported by Institute of Information & Communications Technology Planning & Evaluation (IITP) grant funded by the Korea government(MSIT) (No.2019-0-00075, Artificial Intelligence Graduate School Program (KAIST) and No.2019-0-01396, Development of framework for analyzing, detecting, mitigating of bias in AI model and training data)

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

---

**Algorithm 1** Spread Spurious Attribute

---

1: **Input**: group-labeled set $\mathcal{D}_L$, group-unlabeled set $\mathcal{D}_U$, number of training splits $K$, number of iterations $T$, confidence threshold $\tau_{g_{\min}}$, learning rate $\eta$
2: Split unlabeled dataset $\mathcal{D}_U^{(1)}, \cdots, \mathcal{D}_U^{(K)} \leftarrow \mathcal{D}_U$    {Phase 1: pseudo-labeling}
3: Split labeled dataset $\mathcal{D}_L^\circ, \mathcal{D}_U^\bullet \leftarrow \mathcal{D}_L$
4: **for** $k = 1, \cdots, K$ **do**
5:     Initialize the pseudo-attribute predictor $f_k(x; \theta_k)$
6:     $\mathcal{D}_U^\circ \leftarrow \cup_{\substack{j=1 \\ j \neq k}}^{K} \mathcal{D}_U^{(j)}, \mathcal{D}_U^\bullet \leftarrow \mathcal{D}_U^{(k)}$
7:     **for** $t = 1, \cdots, T$ **do**
8:         Draw a group-labeled mini-batch $\mathcal{B}_L = \{(\tilde{x}^{(b)}, \tilde{y}^{(b)}, \tilde{a}^{(b)})\}_{b=1}^{B}$ from $\mathcal{D}_L^\circ$
9:         Draw a group-unlabeled mini-batch $\mathcal{B}_U = \{(x^{(b)}, y^{(b)})\}_{b=1}^{B}$ from $\mathcal{D}_U^\circ$
10:         Update $\tau_g$ for $g \neq g_{\min}$ as minimum $\tau_g$ satisfying Eq. (7)
11:         Update $\theta_k \leftarrow \eta \nabla_{\theta_k} \left( \sum_{(x,y,a) \in \mathcal{B}_L} \ell_{\sup}(x, y, a) + \sum_{(x,y) \in \mathcal{B}_U} \ell_{\text{unsup}}(x, y) \right)$
12:     **end for**
13:     $\widetilde{\mathcal{D}}_U^{(k)} \leftarrow \left\{ (x, y, f_k(x; \theta_k)) \mid (x, y) \in \mathcal{D}_U^{(k)} \right\}$
14: **end for**
15: $\widetilde{\mathcal{D}}_U \leftarrow \cup_{k=1}^{K} \widetilde{\mathcal{D}}_U^{(k)}$
16: Initialize the robust model $f(x; \theta)$    {Phase 2: robust training}
17: Run Group DRO with group-label estimated training set $\theta \leftarrow \text{GDRO}(f(x; \theta), \widetilde{\mathcal{D}}_U)$

---

## A    EXPERIMENTAL DETAILS

### A.1    SPREAD SPURIOUS ATTRIBUTE PSEUDOCODE

Algorithm 1 provides pseudocode for Spread Spread Attribute combined with Group DRO.

### A.2    DISCUSSIONS ON SPLITTING THE GROUP-LABELED AND GROUP-UNLABELED SETS

Recall that in the pseudo-labeling phase of SSA (Section 4.1, we partition both the group-labeled set and the group-unlabeled set into two subsets:

$$\mathcal{D}_L = \mathcal{D}_L^\circ \cup \mathcal{D}_L^\bullet, \quad \mathcal{D}_U = \mathcal{D}_U^\circ \cup \mathcal{D}_U^\bullet. \tag{10}$$

SSA then trains a spurious attribute predictor based on $\mathcal{D}_L^\circ, \mathcal{D}_U^\circ$, with hyperparameters tuned using $\mathcal{D}_L^\bullet$. The trained model is then used to make predictions on the samples in $\mathcal{D}_U^\bullet$.

Here, it is easy to see that, if we want to make a best prediction on a particular data point $x_\star \in \mathcal{D}_U$, then the optimal split would be the one that uses the largest number of group-unlabeled samples for training the model, i.e.,

$$\mathcal{D}_U^\circ = \mathcal{D}_U \setminus \{x_\star\}, \quad \mathcal{D}_U^\bullet = \{x_\star\}. \tag{11}$$

However, such partitioning requires training $|\mathcal{D}_U|$ different models to label all samples in the group-unlabeled set, which is computationally infeasible. Thus, we propose partitioning the group-unlabeled dataset into $K$ equally-sized subsets $\mathcal{D}_U^{(1)}, \ldots, \mathcal{D}_U^{(K)}$, and run the algorithm $K$ times, using

$$\mathcal{D}_U^\circ = \cup_{\substack{j=1 \\ j \neq i}}^{K} \mathcal{D}_U^{(j)}, \quad \mathcal{D}_U^\bullet = \mathcal{D}_U^{(i)}, \tag{12}$$

for the $i$-th training iteration. For all experiments appearing in this paper, we used $K = 3$ for a simple evaluation; the empirical performance of SSA may improve if we use a larger value of $K$.

Table 8: Group-wise accuracy of spurious attribute predictor in pseudo-labeling phase on CelebA. Results of the worst-performing group are marked in bold.

| Method | Amount of group label used | Non-blond | | Blond | |
|---|---|---|---|---|---|
| | | Female | Male | Female | Male |
| SSA (Ours) | 10% of val. set | 89.5 | 93.9 | 95.1 | **83.6** |
| - Without group-unlabeled set split | | 86.9 | 91.5 | 94.9 | **76.3** |
| SSA (Ours) | 5% of val. set | 83.3 | 90.6 | 92.7 | **75.7** |
| - Without group-unlabeled set split | | 83.4 | 89.5 | 93.9 | **71.7** |

Table 9: Group-wise accuracy of spurious attribute predictor in pseudo-labeling phase on Waterbirds.
Results of the worst-performing group are marked in bold.

| Method | Amount of group label used | Landbird | | Waterbird | |
|---|---|---|---|---|---|
| | | Land | Water | Land | Water |
| SSA (Ours) | 10% of val. set | 92.1 | **96.2** | **94.6** | 93.2 |
| - Without group-unlabeled set split | | 91.3 | **96.2** | **92.9** | 94.3 |
| SSA (Ours) | 5% of val. set | 85.1 | **94.6** | **91.1** | 91.6 |
| - Without group-unlabeled set split | | 85.2 | **94.6** | **91.1** | 92.5 |

For the group-labeled set, we do not require such trick. We simply partitioned the group-labeled into two equally size subsets, and used one for training and another for validation. We kept the split fixed throughout the whole pseudo-labeling phase.

**Discussion.** The splitting procedure aims to mitigate the potential negative effect of "self-confirmation" that may take place in the pseudo-labeling procedure. During the training of pseudo-labeler, SSA utilizes the pseudo-attributes of the training samples whenever the prediction confidence exceeds a certain threshold. In other words, when a sample gains a high-enough confidence, pseudo-labeling may continually strengthen its own predictions, and can be very problematic in group-DRO-like scenarios where we expect a severe group imbalance. Data splitting helps mitigate this effect by separating the samples that we train on and the samples we make final predictions on (as a side note, we also propose group-adaptive thresholds to mitigate the same effect). Empirically, we indeed observe that data splitting helps improve the downstream worst-group performance. Tables 8 and 9 give a comparison of SSA with/without splitting on CelebA and Waterbirds dataset, respectively.

## A.3 TRAINING DETAILS

**Models.** As we briefly discussed in the main text, we use pretrained ResNet-50 and BERT for image and natural language dataset experiments, respectively. For ResNet-50, we use the `torchvision` implementation. For BERT, we use the `huggingface` implementation.

**Hyperparameter Tuning - Overall.** We separately tune the hyperparameters for the pseudo-labeling phase and the robust training phase. For the pseudo-labeling phase, the tuning criterion is the worst-group *spurious label* classification accuracy on $\mathcal{D}_L^\bullet$. For the robust training phase, the tuning criterion is the worst-group prediction accuracy of the trained target attribute classifier on the whole validation set $\mathcal{D}_L$. We fix the threshold $\tau_{g_{\min}}$ for the group with smallest population as 0.95 following (Sohn et al., 2020) in pseudo-labeling phase.

**Hyperparameter Tuning - Image.** For Waterbirds and CelebA, we tuned the learning rate over {1e-3, 1e-4, 1e-5} and $\ell_2$ regularization over {1e-1, 1e-4}. We used SGD optimizer with momentum 0.9 and batch size 64. In pseudo-labeling phase, we train the spurious attribute predictor 1k iterations for Waterbirds and 45k iterations for CelebA.

**Hyperparameter Tuning - Natural Language.** For MultiNLI and CivilComments-WILDS, we did not tune any hyperparameter and follow the details of Liu et al. (2021). For MultiNLI, we trained SSA and its baselines for 5 epochs with default tokenization and dropout, using the batch size 32 and

Table 10: Runtime of pseudo-labeling phase and robust training phase on the datasets we considered.

| Dataset | Waterbirds | CelebA | MultiNLI | CivilComments |
|---|---|---|---|---|
| Pseudo-labeling phase | 3 hrs / split | 2 hrs / split | 4 hrs / split | 7 hrs / split |
| Robust training phase (Group DRO) | 2.3 hrs | 12 hrs | 6.2 hrs | 12.5 hrs |

the initial learning rate of 2e-5. We did not use any $\ell_2$ regularization. For CivilComments-WILDS, we capped the number of tokens per example at 300 and used batch size 8, $\ell_2$ regularization of 1e-2 and initial learning rate of 1e-5. We used AdamW optimizer with gradient clipping for both MultiNLI and CivilComments-WILDS. In pseudo-labeling phase, we train the spurious attribute predictor 30k iterations for both MultiNLI and CivilComments.

**Hyperparameter Tuning - Robust Training.** For the robust model, we use hyperparameters to maximize worst-group accuracy on $\mathcal{D}_{val}$. Following Liu et al. (2021), we use learning rate of 1e-4 and $\ell_2$ regurlarization 1e-1 for Waterbirds, learning rate of 1e-5 and $\ell_2$ regularization 1e-1 for CelebA. We use SGD optimizer with momentum 0.9 and batch size 64 for both Waterbirds and CelebA. For MultiNLI and CivilComments, we use the same configuration as pseudo-labeling phase, except using batch size 16 for CivilComments. In robust training phase, we train the robust model 300 epochs for Waterbirds and CelebA, 10 epochs for MultiNLI, and 5 epochs for CivilComments. We choose the best model based on the model selection criteria described above.

**Baseline Implementation.** For EIIL, we directly take results on Waterbirds and CivilComments from Creager et al. (2021), while the results on CelebA and MultiNLI are new. For environment inference on CelebA, we follow the same procedure as for Waterbirds. We use one epoch trained ERM as a reference classifier. We optimize the EI objective with a learning rate of 0.01 for 20k steps using the Adam optimizer. For environment inference on MultiNLI, we follow the same procedure as for CivilComments-WILDS. We train an ERM model for 5 epochs and choose the reference classifier using the best epoch based on the validation worst-group accuracy. We use the error split heuristic instead of optimizing EI objective as Creager et al. (2021) did. We then train the robust model with the same procedure as we did.

## A.4 RUNTIME ANALYSIS

In Table 10, we provide the time required for the pseudo-labeling phase and the robust training phase on a single Nvidia Titan XP for each dataset. Compared with the vanilla Group DRO, SSA requires an additional pseudo-labeling phase. When we select $K = 3$, the overhead is as small as x0.5 on the CelebA dataset, and does not exceed x4 in the worst case (Waterbird). We make two additional remarks. First, other baseline methods for addressing the lack of full group annotation (e.g., JTT, LfF) also require additional computation and runtime. For instance, JTT requires a preliminary training run to estimate the spurious label. Second, the number of iterations we used for pseudo-labeling has not been optimized for achieving the best runtime-performance tradeoff, and thus can be further improved with a more careful search.

A.5 DETAILS ON SUPERVISED CONTRASTIVE LEARNING

Given an anchor $(x_i, y_i)$, original supervised contrastive loss uses same class samples $(y = y_i)$ as positive samples and different class samples $(y \neq y_i)$ as negative samples to maximize similarity of representation between samples belong to same class. Correct-N-Contrast (CNC; Zhang et al. (2021)) first trains ERM model as JTT and use ERM prediction as a surrogate to ground truth spurious attribute. With obtained ERM prediction, CNC uses supervised contrastive loss to maximize similarity of representation between samples having same target and different ERM prediction, while minimizing similarity of representation between samples having different target and same ERM prediction. Similar to the second stage of CNC, given an anchor $(x_i, y_i, a_i)$, we use samples having same target and different spurious attribute as positive samples $(y^+ = y_i, a^+ \neq a_i)$ and samples having different target and same spurious attribute as negative samples $(y^- \neq y_i, a^- = a_i)$. To be specific, we sample $M$ samples from each group. Given an anchor $(x_i, y_i, a_i)$, we use samples from group $(y_i, a^+)$ for $a^+ \neq a_i$ as a set of positive samples $B^+$ and samples from group $(y^-, a_i)$ for $y^- \neq y_i$ as a set of negative samples $B^-$. In addition to standard cross entropy loss, we minimizes following supervised contrastive loss for each anchor $x_i$:

$$\frac{1}{|B^+|} \sum_{z^+ \in B^+} -\log \frac{\exp(z_i \cdot z^+/\tau)}{\sum_{z \in B^+ \cup B^-} \exp(z_i \cdot z/\tau)}, \tag{13}$$

where $z = g(x)$ is a representation of $x$ and $g : \mathcal{X} \to \mathbb{R}^d$ is an encoder maps $x$ to a representation, $f = h \circ g, \tau > 0$ is a scalar temperature hyperparameter.

We use $M = 16$ for both Waterbirds and CelebA. Except batch size, we follow Zhang et al. (2021) for other hyperparameters. For Waterbirds, we use temperature 0.1, contrastive weight 0.75, SGD optimizer with momentum 0.9, learning rate 1e-4, weight decay 1e-3 and use gradient accumulation to update model parameter every 32 batches. For CelebA, we use temperature 0.05, contrastive weight 0.75, SGD optimizer with momentum 0.9, learning rate 1e-5, weight decay 1e-1 and use gradient accumulation to update model parameter every 32 batches.

A.6 DETAILS ON CLASS-IMBALANCED SEMI-SUPERVISED LEARNING

We consider a classification problem with $K$ classes. In other words, our goal is to train a predictor $\mathcal{X} \to \mathcal{Y}$ with $\mathcal{Y} = \{1, \ldots, K\}$. We assume that we have access to two types of datasets: labeled, and unlabeled. We let

$$\mathcal{D}_L := \{(\tilde{x}_1, \tilde{y}_1), \ldots, (\tilde{x}_m, \tilde{y}_m)\}, \quad \mathcal{D}_U := \{x_1, \ldots, x_n\}. \tag{14}$$

The number of samples in some class $k \in \mathcal{Y}$ will be denoted by $m_k$ and $n_k$, respectively, i.e., $\sum_{k=1}^{K} m_k = n$ and $\sum_{k=1}^{K} m_k = m$. Without loss of generality, we assume that the number of labeled data in each class is ordered in a descending order, i.e.,

$$m_{\mathrm{maj}} =: m_1 \geq m_2 \geq \cdots \geq m_K. \tag{15}$$

We define the *imbalance ratio* of this labeled dataset as the ratio between the sample sizes of the largest class and the smallest class, i.e., $\gamma_{\mathrm{lab}} = m_1/m_K \geq 1$. This quantity is used as a key parameter that controls the degree of class imbalance in the labeled dataset; higher the $\gamma_{\mathrm{lab}}$, more severe the imbalance. To determine the number of samples in other classes, we use an exponential decay function, i.e., $m_k = m_1 \cdot \gamma_{\mathrm{lab}}^{(-k-1)/(K-1)}$. We assume that the same ordering holds for the unlabeled samples, i.e., $n_1 \geq \cdots \geq n_K$, and let define the imbalance ratio as $\gamma_{\mathrm{unlab}} = n_1/n_K$. For simplicity, we let $\gamma_{\mathrm{unlab}} = 1$, i.e., all classes have the same number of unlabeled samples.

To evaluate the classification performance of models trained under the imbalanced dataset, we report two popular metrics: *balanced accuracy* (bACC) and *geometric mean scores* (GM), which are defined by the arithmetic and geometric mean over class-wise sensitivity, respectively. Mean and standard deviation are reported across three random trials, respectively.

