# OpenReview forum: "Spread Spurious Attribute: Improving Worst-group Accuracy with Spurious Attribute Estimation "
_ICLR.cc/2022/Conference — ICLR 2022 Poster_

### Official Review · Reviewer_a6uc · 2021-11-02

**Correctness:** 3
**Technical Novelty And Significance:** 3
**Empirical Novelty And Significance:** 3
**Recommendation:** 6
**Confidence:** 4

**Main Review:**

Strengths：

1. This paper is well-written and well-motivated. The proposed spread spurious attribute method is novel.

2. Comprehensive experimental results demonstrate the effectiveness of the proposed approach.

3. The proposed approach is a general yet effective framework that can be applied to lots of tasks.



**Summary Of The Paper:**

This paper proposes a pseudo-attribute-based algorithm, coined Spread Spurious Attribute (SSA), for improving the worst-group accuracy. The proposed method leverages samples both with and without spurious attribute annotations to train a model to predict the spurious attribute, then uses the pseudo-attribute predicted by the trained model as supervision on the spurious attribute to train a new robust model having minimal worst-group loss. Experimental results on various benchmark datasets show that the algorithm consistently outperforms the baseline methods using the same number of validation samples with spurious attribute annotations.

**Summary Of The Review:**

Overall, this paper proposes a novel and general yet effective framework. The paper is well-motivated and well-written. Moreover, comprehensive experiments have been conducted to demonstrate the effectiveness of the proposed approach.

---

> ### Author Response · Authors · 2021-11-15
> **Response to Reviewer a6uc**
>
> We sincerely appreciate your efforts and insightful comments to improve the manuscript.
> Thank you for your positive comments on our paper. We are glad that you found our proposed method novel and well-supported by comprehensive experiments. If you have any concerns or comments, please let us know.

---

> > ### Comment · Reviewer_a6uc · 2021-11-26
> > **Thanks for your reply**
> >
> > Thanks for your reply. I have read the comments from other reviewers.  Although some parts of the methods are a bit hard to follow, and I have to read twice to understand the details, the overall method is novel and interesting.

---

### Official Review · Reviewer_dgzg · 2021-11-03

**Correctness:** 4
**Technical Novelty And Significance:** 3
**Empirical Novelty And Significance:** 3
**Recommendation:** 6
**Confidence:** 5

**Main Review:**

Strengths:
- The empirical results on worst-group accuracy are strong, even with a small number of attribute-annotated examples.
- The proposed semi-supervised approach seems promising, both for the spurious attribute setting and even the general class-imbalanced SSL setting, as shown in Section 5.4.

Weaknesses:
- The language of training and validation is confusing and imprecise, as the "validation" samples are actually used for training as well. The authors should instead clarify that some training examples have attribute annotations and some do not, and similarly for the validation set.
- The methods description is hard to understand in some parts. Specifically, why are predictions not made in D_{train}^{circ} (as mentioned at the end of Section 4.1)? In principle, the model could make predictions for these, but no clear reason is given why this is not done.
- Related to the previous point, one would expect the proposed algorithm to be very computationally intensive. First, the semi-supervised learning stage will likely be expensive. Next, since this is repeated K=3 times, that makes it even more expensive. (And finally, the Group DRO model still needs to be trained after that.) Computational cost / runtime results are not provided.
- The paper is unclear about some hyperparameter details such as the number of epochs for semi-supervised learning.
- It would be great to better understand why the proposed approach performs so well. Based on Table 3 there is almost no drop in performance until we reach 5% of the original val set size, and even then the drop is small. In this case, on Waterbirds, we would only have 3 annotated examples for worst group for training the spurious attribute predictor (and perhaps fewer if any of these are actually used for validation, rather than training itself?). What is the accuracy of the spurious attribute predictor on the worst group - if it is high, how and why might this be the case (despite the extremely low number of annotations)? If it is low, why does the robust model still attain good worst-group performance?

Questions:
In Table 6, are the SupCon results actually vanilla SupCon or the procedure from Zhang et al. (2021)? Based on the writing, it seems the latter, but the table suggests the former.

**Summary Of The Paper:**

This paper studies the group robustness problem in the setting of spurious correlations, when only a small number of samples have spurious attribute annotations. They present a pseudolabeling algorithm based on FixMatch to pseudolabel the remaining examples, and then use worst-case loss minimization algorithms such as Group DRO to train a more robust model.

**Summary Of The Review:**

Overall, this paper has strong empirical results. The paper could be improved with more precise / clear descriptions of the methods and design decisions, and additional ablation experiments or metrics that help explain how the methods actually perform so well even in the extremely low-annotation setting (in lieu of theoretical analysis, which the paper does not have). Thus, my current score is a weak acceptance.

---

> ### Author Response · Authors · 2021-11-15
> **Response to Reviewer dgzg (2/2)**
>
> **[C4] Hyperparameter details.**
>
> Thank you for pointing this out; we have added hyperparameter details (the number of training iterations) for the semi-supervised learning scheme in Appendix A.2.
>
> *  *  *
>
> **[C5] Why the proposed approach performs so well / Accuracy of the spurious attribute predictor. We provide the prediction accuracy of the spurious attribute predictor.**
>
> We provide the prediction accuracy of the spurious attribute predictor for CelebA in Table 4, and additionally provide for Waterbirds below.
>
> $$ \\begin{array} {lccccc}
> \\hline \\text{Method} & \\text{Group label used} & \\text{Landbird(LB)} & \\text{Landbird(WB)} & \\text{Waterbird(LB)} & \\text{Waterbird(WB)} \\\\
> \\hline \\text{SSA (ours)} & \\text{10\\% of val. set} & 92.1 & 96.2 & 94.6 & 93.2 \\\\
> \\text{SSA (ours)} & \\text{5\\% of val. set} & 85.1 & 94.6 & 91.1 & 91.6 \\\\
> \\hline  \\end{array} $$
>
> From the tables, we observe that the spurious attributes can be accurately predicted (at least 75%~85% when using 5% of val. set) even though the number of group-labeled samples is very small for the worst-group. We think that the reason is that the “spurious correlations” are inherently easier to be learned by deep learning models, as argued by [1,2]. More specifically, the works find that “shortcut” spurious correlations (e.g., male/female in CelebA dataset) are problematic because the deep neural networks prioritizes learning such correlations over learning the desired correlations (e.g., hair color). We think that this “easiness” may be a reason why such spurious attribute predictors can achieve a high accuracy even when learned from a small number of samples.
>
> [1] Geirhos et al., Shortcut learning in deep neural networks. Nature Machine Intelligence, 2020
> [2] Nam et al., Learning from failure: De-biasing classifier from biased classifier. NeurIPS, 2020
>
> *  *  *
>
> **[C6] Clarifications on the SupCon.**
>
> Thank you for pointing this out. The SupCon results in Table 6 indeed follow the Correct-N-Contrast (CNC) procedure from Zhang et al. (2021). We have modified the reference in Table 6 to Zhang et al. (2021). In addition, we also have replaced “modified SupCon” with “CNC” to avoid unnecessary confusion.

---

> ### Author Response · Authors · 2021-11-15
> **Response to Reviewer dgzg (1/2)**
>
> We sincerely appreciate your efforts and insightful comments to improve the manuscript. We respond to each of your comments one-by-one in what follows. In the revised manuscript, we have marked the revisions with “green.”
>
> * * *
>
> **[C1] Confusing use of the terminology “Validation.”**
>
> Thank you for suggesting this. In our initial manuscript, we used the terms “training” and “validation” in accordance with their usages in related work (e.g., [1]). On the other hand, we agree with the reviewer’s point that this choice may potentially cause unnecessary confusion, as we use both training and validation samples for either purposes. To address this, we have changed the terminologies “training” and “validation” to “group-unlabeled” and “group-labeled”, respectively.
>
> [1] Liu et al. "Just train twice: Improving group robustness without training group information." ICML, 2021.
>
> * * *
>
> **[C2] Why do we need to split the training data in Section 4.1?**
>
> The splitting procedure aims to mitigate the potential negative effect of “self-confirmation” that may take place in the pseudo-labeling procedure. During the training of pseudo-labeler, SSA utilizes the pseudo-attributes of the training samples whenever the prediction confidence exceeds a certain threshold. In other words, when a sample gains a high-enough confidence, pseudo-labeling may continually strengthen its own predictions, and can be very problematic in group-DRO-like scenarios where we expect a severe group imbalance. Data splitting helps mitigate this effect by separating the samples that we train on and the samples we make final predictions on (as a side note, we also propose group-adaptive thresholds to mitigate the same effect).
>
> Empirically, we indeed observe that data splitting helps improve spurious attribute predictor accuracy of the worst-performing group. Following tables give a comparison of SSA with/without splitting on CelebA and Waterbirds dataset. We have added a brief explanation for the splitting procedure in Section 4.1, and the tables and a formal discussion to the Appendix A.2.
>
> $$ \\begin{array} {lccccc}
> \\hline \\text{Method} & \\text{Group label used} & \\text{Non-blond(F)} & \\text{Non-blond(M)} & \\text{Blond(F)} & \\text{Blond(M)} \\\\
> \\hline \\text{SSA (ours)} & \\text{10\\% of val. set} & 89.5 & 93.9 & 95.1 & 83.6 \\\\
> \\text{- Without split} &  \\text{10\\% of val. set}  & 86.9 & 91.5 & 94.9 & 76.3 \\\\
> \\hline \\text{SSA (ours)} & \\text{5\\% of val. set} & 83.3 & 90.6 & 92.7 & 75.7 \\\\
> \\text{- Without split} & \\text{5\\% of val. set}  & 83.4 & 89.5 & 93.9 & 71.7 \\\\
> \\hline  \\end{array} $$
>
> $$ \\begin{array} {lccccc}
> \\hline \\text{Method} & \\text{Group label used} & \\text{Landbird(LB)} & \\text{Landbird(WB)} & \\text{Waterbird(LB)} & \\text{Waterbird(WB)} \\\\
> \\hline \\text{SSA (ours)} & \\text{10\\% of val. set} & 92.1 & 96.2 & 94.6 & 93.2 \\\\
> \\text{- Without split} &  \\text{10\\% of val. set}  & 91.3 & 96.2 & 92.9 & 94.3 \\\\
> \\hline \\text{SSA (ours)} & \\text{5\\% of val. set} & 85.1 & 94.6 & 91.1 & 91.6 \\\\
> \\text{- Without split} &  \\text{5\\% of val. set}  & 85.2 & 94.6 & 91.1 & 92.5 \\\\
> \\hline  \\end{array} $$
>
> * * *
>
> **[C3] Runtime Cost of SSA.**
>
> While the primary focus of this work is improving the worst-group performance, we agree to the point that documenting the computational costs may help readers better understand the strength and limitations of the proposed method. Following the reviewer’s suggestion, we have added discussions about the runtime cost of the proposed SSA algorithm in Appendix A.4. For each dataset, the time required for the pseudo-labeling phase and the robust training phase on a single Nvidia Titan XP are as follows.
>
> Waterbird  　　 - Pseudo-Labeling: 3h/split (1k iters)  &nbsp;&nbsp; | Robust Training: 2.3h (300 epochs)
>
> CelebA  　　　　  - Pseudo-Labeling: 2h/split (45k iters) | Robust Training: 12h (300 epochs)
>
> MNLI  　　　　　  - Pseudo-Labeling: 4h/split (30k iters) | Robust Training: 6.2h (10 epochs)
>
> CivilComments - Pseudo-Labeling: 7h/split (30k iters) | Robust Training: 12.5h (5 epochs)
>
> Compared with the vanilla Group DRO, SSA requires an additional pseudo-labeling phase. When we select $K=3$, the overhead is as small as x0.5 on the CelebA dataset, and does not exceed x4 in the worst case (Waterbird). We make two additional remarks. First, other baseline methods for addressing the lack of full group annotation (e.g., JTT, LfF) also require additional computation and runtime. For instance, JTT requires a preliminary training run to estimate the spurious label. Second, the number of iterations we used for pseudo-labeling has not been optimized for achieving the best runtime-performance tradeoff, and thus can be further improved with a more careful search.

---

### Official Review · Reviewer_kQbw · 2021-11-04

**Correctness:** 3
**Technical Novelty And Significance:** 2
**Empirical Novelty And Significance:** 3
**Recommendation:** 8
**Confidence:** 4

**Main Review:**

This paper tackles an important problem and provides a convincing demonstration that current SOTA models are susceptible to reliance on spurious correlation in the training set. I detail key strengths, weaknesses, and additional questions below.

### Strengths
- Reducing the amount of annotations required worst-group loss minimization is an important problem, and this paper presents a simple procedure to help with that problem.
- The empirical portion of the paper seems quite thorough, and the work compares against several recent work in this line.
- This paper includes ablations of different components of the proposed formulation to help understand what each one contributes.
- It seems like the SSA approach might be useful when applied with robust training approaches in general.

### Weaknesses, Concerns, & Additional Questions
- My biggest issue is that the paper has several moving parts in Sec. 4.1 and 4.2, which are really the key contributions of this work. I am not sure how to improve the writing here, but there are several pieces like the different validation and training sets, the two different losses, and then the different group thresholds. Even though it seems clear, it was challenging to be sure that all the pieces fit together.
- I am surprised that the approach works so well given the data size disparity between D_train and D_val. I am not that familiar with the performance of standard unsupervised methods. Is such size disparity usually the case?
- In section 4.2, what does "pseudo-group population ratio of the highly-confident samples" mean?
- It seems like the whole point of the group specific thresholding is to help make sure that the model for learning the spurious/group attributes perform well on the small sample groups. First, I don't think I would call this 'confirmation bias' as this paper does, unless this is a standard term in the unsupervised learning literature. Second, it is unclear to me why the approach used for setting this threshold should be effective. The minimum threshold is initially set on the basis of the size of the smallest group, however, consider the unlikely case where that small group is just n copies of the same sample or somehow homogeneous and low variance. This small group would be more easy to learn than a larger group with larger 'variance' or more 'diversity'. Essentially, I am hoping the authors can clarify why this scheme should be effective on the basis of intuition or a toy example. To be more specific, equation 6 seems critical here. I don't understand why this requirement was chosen, and I worry that the dependence on size of the groups is somewhat unusual.
- To clarify, pseudo-group is $\hat{a}(x)$ for the points in the training set?
- There is an important yet simple experiment that I think is missing. Why can't the authors check the accuracy of their SSA scheme for a dataset where we have the ground truth spurious attributes like the waterbirds and the other datasets that they use? Is this what Table 4 is showing? If yes, then it might help to clarify the caption and add more clarification.

Post Rebuttal
Satisfied with the author response, so recommending that the accept.

**Summary Of The Paper:**

This paper presents a technique, spread spurious attribute, for inferring the group annotation for training samples in a dataset. The inferred group information is then used as part of a group DRO minimization scheme or some other worst case scheme. The key insight in this work is to use a small validation set (1-5 percent) of data that has annotations to learn annotations for the entire training set via pseudo-labelling. The labelling scheme presented uses different thresholds for the different spurious groups. This new dataset is then used as part of a new DRO pipeline.

**Summary Of The Review:**

Overall, this paper conducts thorough empirical analysis of the SSA approach, which can be incorporated with robust training methods. I recommend a weak accept because there are portions of the scheme that I think needs better justification.

---

> ### Author Response · Authors · 2021-11-15
> **Response to Reviewer kQbw (2/2)**
>
> **[C6] Clarification on pseudo-group.**
>
> For a training data sample $x$, the pseudo-group is defined as the pair $(y, \hat{a}(x))$, where y is the label and the $\hat{a}(x)$ denotes the _pseudo-attribute_ of the points in the training set.
>
> *  *  *
>
> **[C7] SSA’s accuracy on predicting ground truth spurious attributes.**
>
> As you mentioned, Table 4 is giving SSA’s prediction accuracy (recall) of the ground truth spurious attributes for each group. We have clarified this in the revised manuscript by modifying the paragraph header “Performance comparison” to “Accuracy of the spurious attribute predictor” and the table caption.

---

> > ### Comment · Reviewer_kQbw · 2021-11-30
> > **Clarified my concerns**
> >
> > Thank you for the response, most of my concerns have now been addressed.

---

> ### Author Response · Authors · 2021-11-15
> **Response to Reviewer kQbw (1/2)**
>
>  We sincerely appreciate your efforts and insightful comments to improve the manuscript. We respond to each of your comments one-by-one in what follows. In the revised manuscript, we have marked the revisions with “green.”
>
> *  *  *
>
> **[C1] Moving parts in Section 4.1 and 4.2.**
>
> Thank you for pointing this out. Following your suggestion, we have edited and augmented section 4 to further clarify what SSA is doing, including the added high-level descriptions and some changes in terminology. We have also added a pseudocode of our algorithm in Appendix A.1 to illustrate the overall training procedure.
>
> *  *  *
>
> **[C2] Effectiveness of SSA given the data size disparity.**
>
> In fact, having such a large sample size disparity between the unlabeled and labeled datasets is quite common in the semi-supervised learning literature. For instance, Kim et al. [1] and Oh et al. [2] use a validation set composed of only 10 samples from each class. Given this fact, there are two additional reasons why SSA may require even fewer labeled samples than typical semi-supervised learning scenarios. First, the goal of SSA is not to recover the spurious attributes perfectly, but to use potentially-noisy pseudo-labels on spurious attributes as a basis to perform group DRO (which requires less information, presumably). Second, the spurious correlations are typically considered “easy to be learned” by neural networks than the desired correlations, which makes such correlations a harmful shortcuts (see, e.g., [3,4])
>
> [1] Kim et al., "Distribution aligning refinery of pseudo-label for imbalanced semi-supervised learning." NeurIPS, 2020 \
> [2] Oh et al., "Distribution-Aware Semantics-Oriented Pseudo-label for Imbalanced Semi-Supervised Learning." arXiv preprint, 2021 \
> [3] Geirhos et al., Shortcut learning in deep neural networks. Nature Machine Intelligence, 2020 \
> [4] Nam et al., Learning from failure: De-biasing classifier from biased classifier. NeurIPS, 2020
>
> *  *  *
>
> **[C3] Clarifying the term “pseudo-group population ratio …”.**
>
> The term “highly-confident samples” denotes the training samples in $\mathcal{D}_{\textrm{train}}^\circ$ that whose prediction confidence of the spurious attribute predictor exceeds the group-specific threshold $\tau$ (and thus being actively used for updating the weights of the predictor). The phrase “pseudo-group population ratio” denotes the ratio of the predicted pseudo-group identities (pseudo-attribute + label) among such samples without spurious attribute annotations. In an ideal case, we may want the number of samples used to update the spurious attribute predictor to grow uniformly for each group, i.e., the pseudo-group population ratio of highly-confident samples is even. We have clarified this point in the revised manuscript.
>
> *  *  *
>
> **[C4] On the term “Confirmation bias.”**
>
> We see the reviewer’s concern. We have replaced the “confirmation bias” with the phrase “pseudo-labels being biased towards the majority group” in the revised manuscript.
>
>
> *  *  *
>
> **[C5] Determining threshold on the basis of group size.**
>
> As the reviewer mentioned, we decide adaptive thresholds to make “number of samples used for spurious attribute predictor update” to be even for each group (Eq. 6). This design essentially shares a similar spirit with the “majority group subsampling,” which is known to be an effective strategy to mitigate spurious correlation, especially in the overparameterized regime [1]. On the one hand, it is true that thresholds actively accounting for the diversity/variance of each subgroup may enjoy advantages over the cardinality-based thresholds under the presence of severe variance disparity among group-wise data distributions (like the scenario the reviewer described); indeed, in the class imbalance literature, several recent works use such variance-based measures to better resolve the class imbalance problem [2,3]. On the other hand, however, such techniques are difficult to be directly combined with our method, as they typically require a variance estimation step which may be extremely noisy when we have a very small number of labeled samples for the smallest group (like the scenario we consider). We chose to use the simple size-based criteria, as it is a simple and effective strategy that does not require any sophisticated variance-estimation.
>
> Nevertheless, we agree with the reviewer’s point that ideally the algorithm should be able to account for the intragroup diversity of each subgroup. We think this would be a very interesting future direction, and believe that our work will serve as a strong baseline for the follow-up works.
>
> [1] Sagawa et al. "An investigation of why overparameterization exacerbates spurious correlations.” ICML, 2020 \
> [2] Khan et al. "Striking the right balance with uncertainty." CVPR, 2019 \
> [3] Liu et al. "Deep representation learning on long-tailed data: A learnable embedding augmentation perspective." CVPR, 2020

---

### Official Review · Reviewer_qZ2x · 2021-11-06

**Correctness:** 3
**Technical Novelty And Significance:** 2
**Empirical Novelty And Significance:** 2
**Recommendation:** 6
**Confidence:** 3

**Main Review:**

## Long summary
Previous methods try to identify and upweight samples from minority groups (spurious attribute joint with class label). Then a validation set of annotated samples are used to tune hyperparameters. The performance of such models are very sensitive to these hyperparameters/validation set, and therefore, fail to perform comparably with methods that use the spurious attribute. This work aims to resolve this issue to achieve performance similar to methods with annotations.

SSA first does pseudo labeling and then does robust training. It trains a domain index predictor and uses a threshold to solve the confirmation bias issue by balancing each group. Then pseudo labels are used on the training set. Then group DRO is used for robust training.

The confidence of the majority group increases faster than the minority group, which is even more severe when the label is pseudo label. This is so-called confirmation bias.

In experiments, it uses the datasets widely used in this field: Waterbirds, CelebA, MultiNLI and CivilComments-WILDS. They show SSA improve the worst group accuracy significantly over existing methods without spurious attribute even with much less valiation data. They also analyze the influence of the validation set size. Finally, they analyze the pseudo labeling process of SSA to verify the adaptive thresholding is useful. They also show SSA works with supervised contrastive learning loss and can be useful in semi supervised learning with imbalanced classes.


## Strength:

1. Very comprehensive experiments.
2. Relatively simple and reproducible method.

## Weakness:

1. Lack of very strong theoretical justification. It is hard to have a straightforward interpretation of why this works, especially based on pseudo labels.
2. Some experiments seem not relevant to the problem this paper aims to solve.

## details and questions
1. Table 6 is kind of redundant since most of it overlaps with Table 1.
2. It is not clear why the supervised contrastive learning part is needed in this work. The same to the semi-supervised learning experiments.
3. How are the supervised and unsupervised loss combined (eq.3)?
4. Why do we need to split the training data in 4.1? They don't have spurious attributes anyway and the method only infers them on training data.
5. I believe it can also be compared with EIIL [1].

[1] Creager, Elliot, Jörn-Henrik Jacobsen, and Richard Zemel. "Environment inference for invariant learning." International Conference on Machine Learning. PMLR, 2021.

**Summary Of The Paper:**

This work focuses on the worst group optimization (distributional robust optimization) problem with few spurious attributes are available as a validation set. They aim to achieve similar performance with methods that use spurious attributes. The main contribution is an adaptive thresholding method to ensure balanced sample from each group. Experiments show the proposed pseudo labeling and thresholding are effective in various tasks.


**Summary Of The Review:**

## Strength:

1. Very comprehensive experiments.
2. Relatively simple and reproducible method.

## Weakness:

1. Lack of very strong theoretical justification. It is hard to have a straightforward interpretation of why this works, especially based on pseudo labels.
2. Some experiments seem not relevant to the problem this paper aims to solve.

---

> ### Author Response · Authors · 2021-11-15
> **Response to Reviewer qZ2x (2/2)**
>
> **[C5] Why do we need to split the training data in Section 4.1?**
>
> The splitting procedure aims to mitigate the potential negative effect of “self-confirmation” that may take place in the pseudo-labeling procedure. During the training of pseudo-labeler, SSA utilizes the pseudo-attributes of the training samples whenever the prediction confidence exceeds a certain threshold. In other words, when a sample gains a high-enough confidence, pseudo-labeling may continually strengthen its own predictions, and can be very problematic in group-DRO-like scenarios where we expect a severe group imbalance. Data splitting helps mitigate this effect by separating the samples that we train on and the samples we make final predictions on (as a side note, we also propose group-adaptive thresholds to mitigate the same effect).
>
> Empirically, we indeed observe that data splitting helps improve spurious attribute predictor accuracy of the worst-performing group. Following tables give a comparison of SSA with/without splitting on CelebA and Waterbirds dataset. The tables and a formal discussion are added to the Appendix A.2.
>
> $$ \\begin{array} {lccccc}
> \\hline \\text{Method} & \\text{Group label used} & \\text{Non-blond(F)} & \\text{Non-blond(M)} & \\text{Blond(F)} & \\text{Blond(M)} \\\\
> \\hline \\text{SSA (ours)} & \\text{10\\% of val. set} & 89.5 & 93.9 & 95.1 & 83.6 \\\\
> \\text{- Without split} &  \\text{10\\% of val. set}  & 86.9 & 91.5 & 94.9 & 76.3 \\\\
> \\hline \\text{SSA (ours)} & \\text{5\\% of val. set} & 83.3 & 90.6 & 92.7 & 75.7 \\\\
> \\text{- Without split} & \\text{5\\% of val. set}  & 83.4 & 89.5 & 93.9 & 71.7 \\\\
> \\hline  \\end{array} $$
>
> $$ \\begin{array} {lccccc}
> \\hline \\text{Method} & \\text{Group label used} & \\text{Landbird(LB)} & \\text{Landbird(WB)} & \\text{Waterbird(LB)} & \\text{Waterbird(WB)} \\\\
> \\hline \\text{SSA (ours)} & \\text{10\\% of val. set} & 92.1 & 96.2 & 94.6 & 93.2 \\\\
> \\text{- Without split} &  \\text{10\\% of val. set}  & 91.3 & 96.2 & 92.9 & 94.3 \\\\
> \\hline \\text{SSA (ours)} & \\text{5\\% of val. set} & 85.1 & 94.6 & 91.1 & 91.6 \\\\
> \\text{- Without split} &  \\text{5\\% of val. set}  & 85.2 & 94.6 & 91.1 & 92.5 \\\\
> \\hline  \\end{array} $$
>
> *  *  *
>
> **[C6] Comparison with EIIL.**
>
> In our initial version, we did not compare with EIIL [1] as the goal of EIIL is slightly different from our paper (and the baseline methods)---EIIL aims for domain-invariance, while SSA aims for improving the worst-group performance without an explicit consideration for invariance. In fact, this distinction has been readily made clear in Table 1 of [1].
>
> On the other hand, as EIIL also reports an improvement in terms of the worst-group accuracy, we agree that adding an experimental comparison with EIIL will make our paper stronger. To this end, we have added experimental results on EIIL in Tables 1 and 2 of the revised manuscript (the experimental settings are added to Appendix A.3); the results on Waterbirds and CivilComments are directly taken from [1], while the results on CelebA and MultiNLI are new.
>
> The following table gives a quick comparison of the added results. We observe that the worst-group accuracy of SSA is higher than those of JTT and EIIL throughout all datasets; EIIL underperforms JTT in all datasets except for CelebA. The results reassures the strength of the proposed SSA.
>
> $$ \\begin{array} {lccccccc}
> \\hline
> & \\text{Waterbirds} & & \\text{CelebA} & & \\text{MultiNLI} & & \\text{CivilComments} & \\\\
> \\text{Method} & \\text{Avg.} & \\text{Worst-group} & \\text{Avg.} & \\text{Worst-group} & \\text{Avg.} & \\text{Worst-group} & \\text{Avg.} & \\text{Worst-group} \\\\
> \\hline \\text{EIIL} & 96.9 & 78.7 & 91.9 & 83.3 & 79.4 & 70.9 & 90.5 & 67.0 \\\\
> \\text{JTT} & 93.3 & 86.7 & 88.0 & 81.1 & 78.6 & 72.6 & 91.1 & 69.3 \\\\
> \\text{SSA (Ours)} & 92.2 & 89.0 & 92.8 & 89.8 & 79.9 & 76.6 & 88.2 & 69.9 \\\\
> \\hline  \\end{array} $$
>
>
> [1] Creager, Elliot, Jörn-Henrik Jacobsen, and Richard Zemel. "Environment inference for invariant learning." ICML 2021.

---

> > ### Comment · Reviewer_qZ2x · 2021-11-16
> > **Response to authors**
> >
> > I really appreciate the response with the many new experimental results.
> > Thanks for providing the insights in [C1].
> > We actually can observe in [C5], SSA without split outperforms SSA in Waterbird while it is not as good as SSA in CelebA.
> > The comparison with EIIL verifies SSA is better in many cases (worst-group performance).

---

> ### Author Response · Authors · 2021-11-15
> **Response to Reviwer qZ2x (1/2)**
>
> We sincerely appreciate your efforts and insightful comments to improve the manuscript. We respond to each of your comments one-by-one in what follows. In the revised manuscript, we have marked the revisions with “green.”
>
> *  *  *
>
> **[C1] Justification / Interpretation of the proposed pseudo-labeling-based method.**
>
> In a nutshell, SSA first estimates the spurious attributes of the samples without group annotation, and uses the recovered group information to find a predictor that performs well on all groups. For the first step (subgroup estimation), SSA uses “pseudo-labeling,” which is known to be one of the most powerful methods to estimate unknown annotations of the samples in the semi-supervised learning literature---the effectiveness of the pseudo-labeling strategy has been validated both empirically [1,2] and theoretically [3,4]. The strategy is known to be especially effective when the subgroup population is severely imbalanced [5,6], which is a common scenario in the group DRO literature that we consider. In the proposed SSA, we use the (improved) pseudo-labeling technique to approximately recover the “spurious attribute annotations” of the training samples with known target attribute annotations, unlike a typical semi-supervised learning scenario where we recover the target attribute annotations without any auxiliary annotations. Using the recovered group annotations, we leverage known algorithms for training group distributionally robust predictors.
>
> At a high level, SSA brings a (well established pseudo-labeling) technique in another field (semi-supervised learning) to solve our problem with a novel modification. Hence, why SSA works is analogous to why pseudo-labeling techniques work for semi-supervised learning.
>
> [1] Sohn et al., “FixMatch: Simplifying Semi-supervised Learning with Consistency and Confidence.” NeurIPS 2020. \
> [2] Zhang et al., “FlexMatch: Boosting Semi-supervised Learning with Curriculum Pseudo Labeling.” NeurIPS 2021. \
> [3] Oymak et al. “A Theoretical Characterization of Semi-supervised Learning with Self-training for Gaussian Mixture Models.” AISTATS 2021. \
> [4] He et al. “Information-theoretic generalization bounds for iterative semi-supervised learning.” arXiv 2021. \
> [5] Kim et al., "Distribution aligning refinery of pseudo-label for imbalanced semi-supervised learning." NeurIPS 2020. \
> [6] Wei et al., "Crest: A class-rebalancing self-training framework for imbalanced semi-supervised learning."  CVPR 2021.
>
> *  *  *
>
> **[C2] Overlap of Table 1 and Table 6.**
>
> Experiments in Table 6 aim to show that our method can be generally applicable to various robust training methods by replacing Group DRO with SupCon (CNC in the revised draft). Following your suggestion, we have removed results of unnecessary baseline (JTT) to reduce redundancy in Table 6. To emphasize general applicability of SSA, we redesigned the table and additionally reported the amount of gap closed by SSA between ERM and robust training methods.
>
> *  *  *
>
> **[C3] Why is the supervised contrastive learning experiment needed?**
>
> Recall that the proposed SSA framework consists of two phases: Pseudo-labeling (estimating spurious attributes), and robust training (using estimated attributes to train robust predictors). While we use group DRO as a default subroutine for robust training, the benefit of the proposed framework is not limited to this specific choice of robust training algorithm. Indeed, our experiments with supervised contrastive learning demonstrates that SSA combined with SupCon achieves a performance close to that of “SupCon using full supervision on spurious attributes,” using only a small number of samples with spurious attribute annotations. This result implies that SSA can potentially be combined with an even broader class of robust training algorithms. We clarified this point at the beginning of Section 5.3 in the revision.
>
> *  *  *
>
> **[C4] How are supervised and unsupervised losses combined?**
>
> The final combined loss is sum of two losses:
> $\\mathbb{E}\_{\\mathcal{D}\_L^\\circ} [\\ell\_\\textrm{sup}(x, y, a)] + \\mathbb{E}\_{\\mathcal{D}\_U^\\circ} [\\ell\_\\textrm{unsup}(x, y)]$. \
> In the revised manuscript, we explicitly give this combined loss in Eq. (4) of Section 4.1, and additionally provide a pseudo-code of SSA (Appendix A.1) to further clarify.

---

### Author Response · Authors · 2021-11-15
**General Response**

Dear reviewers and AC,

We sincerely appreciate your valuable time and effort spent reviewing our manuscript. We are glad to find that all reviewers are positive on our paper, that our paper tackles an important and well-motivated problem (reviewer kQbw, a6uc), and provide a novel (a6uc) and generally applicable algorithm (kQbw, a6uc), showing a strong empirical results on comprehensively designed experiments (all reviewers).

We appreciate your constructive feedback on our manuscript. In response to the comments, we have carefully revised and enhanced the manuscript, including the following notable changes:
- Added empirical comparison with EIIL (Table 1,2).
- Clearer description and added pseudocode of the proposed SSA (Section 4.1~4.2, and Appendix A.1).
- Extended discussions and ablation experiments on “training data splitting” during the pseudo-labeling phase (Appendix A.2)
- Runtime cost analysis (Appendix A.4).
- Clearer description of experimental results (Section 4.2. and Table 4 caption).
- Additional discussions on the applicability of SSA to other robust training methods (Section 5.3, Table 6).
- Change of terminologies for clarity, including “validation set” and “confirmation bias” (whole draft).
In the revised manuscript, we have marked the revisions with “green.”

We sincerely believe that these updates may help us better deliver the benefits of the proposed algorithm SSA to the ICLR community.

Thank you very much,

Authors.

---

### Author Response · Authors · 2021-11-19
**A gentle reminder**

Dear Reviewers,

Thank you for your time and efforts in reviewing our paper.

We kindly remind that the discussion period will end soon (in a few days). We believe that we sincerely and successfully address your concerns/questions/misunderstandings/suggestions, with the results of the supporting experiments.

If you have any further concerns or questions, please do not hesitate to let us know.

Thank you very much!

Authors

---

### Decision · Program_Chairs · 2022-01-20

**Decision:**

Accept (Poster)

**Comment:**

This paper presents a new method to decrease the supervision cost for learning spurious attributes using worst-group loss minimization. Their method uses samples both with and without spurious attribute annotations to train a model to predict the spurious attribute, then use the pseudo-attribute predicted by the trained model as supervision on the spurious attribute to train a new robust model having minimal worst-group loss. The experiments show promising results in this domain for reducing annotation cost.

The reviewers vote to accept the paper, and some of them increased their scores during the discussions since the authors have addressed their concerns.